# ANOVA Design for the Optimization of TiO_2_ Coating on Polyether Sulfone Membranes

**DOI:** 10.3390/molecules24162924

**Published:** 2019-08-12

**Authors:** Yasin Orooji, Ehsan Ghasali, Nahid Emami, Fatemeh Noorisafa, Amir Razmjou

**Affiliations:** 1College of Materials Science and Engineering, Nanjing Forestry University, Nanjing 210037, China; 2Department of Biotechnology, Faculty of Advanced Sciences and Technologies, University of Isfahan, Isfahan 73441-81746, Iran; 3UNESCO Centre for Membrane Science and Technology, School of Chemical Science and Engineering, University of New South Wales, Sydney 2052, Australia

**Keywords:** Nanocomposite membrane, ANOVA analysis, Polyether sulfone, antifouling, coating

## Abstract

There have been developments in the optimization of polyethersulfone (PES) membranes, to provide antifouling and mechanically stable surfaces which are vital to water purification applications. There is a variety of approaches to prepare nanocomposite PES membranes. However, an optimized condition for making such membranes is in high demand. Using experimental design and statistical analysis (one-half fractional factorial design), this study investigates the effect of different parameters featured in the fabrication of membranes, as well as on the performance of a nanocomposite PES/TiO_2_ membrane. The optimized parameters obtained in this study are: exposure time of 60 s, immersion time above 10 h, glycerol time of 4 h, and a nonsolvent volumetric ratio (isopropanol/water) of 30/70 for PES and dimethylacetamide (PES-DMAc) membrane and 70/30 for PES and N-methyl-2-pyrrolidone (PES-NMP) membrane. A comparison of the contributory factors for different templating agents along with a nanocomposite membrane control revealed that F127 triblock copolymer resulted in an excellent antifouling membrane with a higher bovine serum albumin rejection and flux recovery of 83.33%.

## 1. Introduction

Membranes are used for a variety of applications including food packaging [1], rechargeable batteries [2], hydrogen production [3], gas separation [4], biomedical applications [5], biodesalination [6], wastewater treatment [7] and fuel cells [8], etc. Polyether sulfone (PES) has been extensively applied for membrane fabrication due to its high thermal and chemical stability and blood compatibility [1]. Mixed matrix membranes have attracted great attention in the last decades [9,10]. TiO_2_ have been explored to treat wastewaters and we have already demonstrated that the incorporation of TiO_2_ nanoparticles can significantly enhance the performance of our membranes [11,12,13]. Although it has been reported that the concentration of TiO_2_ nanoparticles within PES and polyvinylidene fluoride (PVDF) matrices using the phase inversion approach is about 2 wt.%, there is still a discrepancy in the literature with respect to TiO_2_ coating on PES membranes [14]. Phase inversion is one of the most popular methods for membrane fabrication. The conditions of membrane formation in this technique have a significant influence on the morphology, property, and performance of the membrane. It can change the structure of the membrane from a finger- to a sponge-like structure. Phase separation mechanism is based on a ternary system consisting of a solvent (*N*-methyl-2-pyrrolidone/dimethylacetamide, NMP/DMAc), a nonsolvent (water/isopropanol), and a polymer (PES) [10]. In this process, a polymer is cast on a plate as support and, after evaporation of the volatile solvent, it immerses in a coagulation bath of nonsolvent.

The membrane is formed during the exchange of solvent and nonsolvent. The optimization of the parameters affecting membrane fabrication as a support and nanoparticle coating for making nanocomposites seems to be a necessary step, not only for saving time and effort but also for reducing variability in the process and providing a baseline for future work. The literature features many research works which have investigated variable conditions which include the temperature and composition of a casting solution (concentration, additive, and solvent) [11,12,13], solvent evaporation [13,14], temperature of nonsolvent [15], temperature of dope [16], and additional materials in the nonsolvent during phase inversion. Although many membrane preparation parameters that affect membrane structures have been investigated in the literature [17], few studies have been conducted on the optimization of these parameters through statistical analysis.

This research considers the preparation of PES membranes through the manipulation of nanocomposite membrane fabrication variables, including air exposure time, immersion time, solvent ratio (ratio of less volatile solvent (NMP) to more volatile solvent DMAc), glycerol time, and nonsolvent ratio (isopropanol to water) that effect the support structure and additive materials proportion (titanium dioxide), and different templating agents poly(ethylene glycol) (PEG), Pluronic F127 (F127), PEG 60 wt.%– PDMS 40 wt.% copolymer (IM22, PDMS stands for polydimethylsiloxane) which effectively improve the permeability, self-cleaning, and antifouling properties of a nanocomposite membrane. The sol-gel method employed for the synthesis of a nanocomposite membrane is a technique used to increase the chemical, mechanical, and thermal stability of polymeric membranes without a decrease in their performance [18]. Because the performance of membranes depends on the properties of their surfaces, membrane surface modification is one of the most efficient methods to mitigate surface fouling. Generally, membranes which have a hydrophilic surface and neutral charge are less susceptible to protein adsorption and membrane fouling. Titania nanoparticles are well known because they possess an excellent combination of antibacterial and hydrophilic properties [19] which have recently been utilized to modify antifouling properties of membranes [20,21] by forming a hierarchical structure on surfaces.

In this study, coating layers that provide optimum preparation conditions for the phase separation process are studied. This study is based on the performance of PES membranes in water treatment conducted by investigating different parameters for membrane fabrication and various parameters for the fabrication of a nanocomposite PES/TiO_2_ membrane. Our results show that the fraction factorial design along with two-way analysis of variance (ANOVA) could effectively be used to identify important parameters and their main effects, as well as interactions, on the preparation of PES membranes. It is reasonable that the membrane casting solvent affects the surface properties of the membrane [1,22]. The second aim is the optimization of sol-gel parameters to achieve desirable permeability and performance of a nanocomposite membrane. The effects of the concentration of TiO_2_ nanoparticles and that of different types of template agents are investigated. The F127 triblock copolymer acts as a structure-directing agent in the sol-gel process. The IM22 is a hydrophilic polymer which acts as a templating agent. Scanning electron microscope (SEM) and UV-visible spectrophotometer are applied for measuring the skin layer thickness and rejection, respectively.

## 2. Experimental Section

### 2.1. Material

Polyethersulfone (PES; Ultrason E6020P, 58 kDa, BASF Co. Ltd., Ludwigshafen, Germany) was purchased for the membrane matrix preparation. Scharlav Chemie S.A. provided *N*-methyl-2-pyrrolidone (NMP) and dimethylacetamide (DMAc) as a solvent with a molecular weight of 99.13 and 87.12, respectively. Isopropanol as a nonsolvent, ethanol (anhydrous), and glycerol for post-treatment were purchased from Ajax Finechem Pty Ltd. Polyvinylpyrrolidone (PVP; 40 kDa), Pluronic F127, IM22, PEG, and, TTIP 97% were obtained from Sigma-Aldrich. Bovine serum albumin (BSA; 67 kDa) was obtained from MorEgate Biotech. All the other chemicals were of the highest purity and were used without further purification. Materials and additional details are listed in Appendix A.

### 2.2. Fractional Factorial Design

Factorial designs, which allow multiple factors to be investigated at the same time, are an efficient method that could be employed in the study of membrane fabrication. Factorial designs enable more accurate identification of interactions among factors and allow the effects of one factor to be estimated at several levels of other factors studied. In comparison to one-factor-at-a-time experiments, factorial designs reduce the number of experimental runs required to determine the effect of changing one of the factors [23]. A two-level factorial design, low (−) and high (+), was employed in this study as an upper and lower limit of each factor. Therefore, the total number of combinations was 32 (25). The pure water flux, 3 replications, and membrane thickness were considered as responses of each combination. Since the total runs were too numerous (3 × 32 for fluxes and 32 for thicknesses, 128 overall), a one-half fractional factorial design was adopted to reduce the experiments which were 16 (25-1) plus one center point. In a two-level factorial design, the responses are assumed to be approximately linear over the range of the factor levels chosen. Therefore, a potential concern of the two-level factorial design is the assumption of linearity in the factors. Of course, perfect linearity is unnecessary, and the two-level design works quite well even when the linearity assumption holds only approximately [24]. However, adding center points protects curvature. Here, one center point was considered to check the likelihood of curvature.

### 2.3. Membrane Preparation

Awet phase inversion technique was carried out for preparation of the membranes. First, PVP (4 wt.%) was dissolved to the desired ratio of NMP/DMAc (100%, 50%, and 0%). The solution was heated to 60 °C. Secondly, 16% of PES was added under stirring condition until a homogenous solution was obtained. The resulting solution was kept for 4 h for degassing. Casting solutions were cast by a casting machine (Sheen 1133N automatic film applicator) at a speed of 60 mm per second at room temperature and 50% humidity (Appendix A). To investigate the effect of air exposure time, the cast polymers were pre-evaporated for 60, 30, and 0 s and then, the polymers were immersed in a mixture of water and isopropanol (100%, 50%, and 0% by volume). The membranes were immersed in a coagulation bath for different times (1, 8.5, and 16 h) to determine the influence of immersion time. Ultimately, the membranes were kept in glycerol for 0, 2, and 4 h to assess the effect of immersion time in glycerol on the structure and performance of the membranes. The membranes were dried by keeping them between tissue papers.

### 2.4. Effect of Template Agent and TTIP Concentration

To determining the effect of the templating agents on the performance of the membranes, F127, IM22 (as a surfactant), and polyethylene glycol with different concentrations were used (Appendix A). On the basis of our previously reported approach [25], TiO_2_ nanoparticle was synthesized via the sol-gel method by mixing 0.504 g Pluronic F127 in anhydrous ethanol at room temperature. A second solution was prepared by mixing anhydrous ethanol with 2,4-pentanedione, perchloric acid, and titanium (IV) isopropoxide (TTIP), and ultrapure Milli-Q water also at room temperature. The solutions above were stirred separately for 1 h with a magnetic stirrer (IKA RCT Basic, In Vitro Technologies). Subsequently, they were mixed and stirred for another hour. A stable sol formed with a pH of 1.2.

The molar ratios of each component in the resulting sol were TTIP:Pluronic F127:2,4-pentanedione: HClO_4_:H_2_O:ethanol = 1:0.004:0.5:0.5:0.45:4.76. To investigate the effect of TiO_2_ on the membrane performance, the amount of TTIP in the sol solution was decreased from 1 to 0.5, 0.2, and 0.12 molar ratios. Then, the TiO_2_ nanoparticles were coated onto the PES membranes (16% PES, 4% PVP, and NMP solvent with 150 µm thickness) using a low-temperature hydrothermal (LTH) process through dip coating. Afterward, the coated membranes were dried in a vacuum oven at 120 °C for 16 h and placed in Milli-Q water and heated at 90 °C, and then rinsed three times for further characterization.

### 2.5. Membrane Characterization

The morphology of membranes was characterized by field emission scanning electron microscopy (FESEM, Hitachi 4500II). Samples were frozen in liquid nitrogen then fractured for cross-section imaging. The samples were sputtered with chromium to become electrically conductive. The flux measurement of pure water was done in a dead-end cell at 1 bar and room temperature. It should be noted that the fluxes decreased gradually and then became constant after about 20 min. The BSA (0.5 wt.%) filtration tests were also done in a dead-end cell. The flux was measured at 20 min, 1 h, and 2 h. The permeate samples were also taken at the same times for measuring their rejection by UV spectrophotometer.

## 3. Results and Discussions

### 3.1. One-Half Fractional Factorial Design

Simulation and analysis of experimental data were systematically conducted following a one-half fractional factorial design using Minitab Software to examine the effects. The interactions of solvent ratio (A), exposure time (B), immersion time (C), glycerol time (D), and nonsolvent ratio (E) on fluxes of polyethersulfone membranes within an empirically selected range of high (+) and low (−) levels are summarized in Table 1 and Table 2.

A set of seventeen runs representing every combination of the four factors at each level was performed in a random order corresponding to responses (flux, a total of three replicates, and top layer thickness). Appendix A shows the typical probability plot of the effects and their interactions on fluxes. Points that do not fall near the line usually signal important effects. The important effects are more significant than the unimportant ones and are generally farther from the fitted line. The unimportant effects tend to be smaller and are centered around zero. Here, the logarithm of the average flux was considered as a response, because it has a better match with the normal distribution model. Therefore, the main effects were A and D as well as AE and DE as significant interactions. As can be seen, although the impact of the nonsolvent ratio (E) was insignificant, its interaction with other factors such as A and D was significant. Analysis of variance (ANOVA) conducted for response assured that these effects identified from the normal probability plots were statistically significant at the 0.05 confidence level.

The plots in Appendix A represent the main effects of two significant factors, i.e., A and D, plus E on the flux. The negative slope of the largest effect, A, suggests that as the solvent ratio increased from the low level (0) to the high level (100), fluxes decreased to a low level. This means that the membrane prepared by DMAc had higher fluxes due to the greater porosity of the membrane, including a dense layer in the upper part and finger-like pores in the lower part of the membrane (to be discussed further in detail). This effect is also observed in other works [26]. The effect of D is positive, implying that glycerol as a pore-filling agent could effectively preserve the membrane pores and prevent pore collapse after drying. Because the slope of effect E (0.001) is close to zero, the main effect of E itself is insignificant, while its interaction is not meaningless.

The interaction effect of nonsolvent ratio and the solvent ratio (AE) on an average flux are shown in Appendix A. The graph indicates that the effect of nonsolvent ratio on factor A at a high level is more prominent than at a low level. This means that the membranes prepared with NMP were more dependent on the nonsolvent ratio (the more isopropanol in the coagulation bath, the less the flux). As shown in Appendix A, although an increase in isopropanol results in a decrease in flux for NMP, avoiding this substance in the coagulation bath is not recommended. This is due to the distance of the center point from the NMP line. The center point is much farther away from the NMP line than DMAc, which leads to the existence of the curvature. The two-way ANOVA was applied later to investigate the potential optimum possibilities.

The AD interaction plot (Appendix A) shows that if the glycerol time (D) is at a low level, the solvent ratio (A) has a relatively small effect. However, if the glycerol time is at a high level, the solvent ratio has a large effect. This implies that the presence of glycerol in the pores prevents them from collapsing after drying. The plot also indicates that the pores in the membrane prepared with DMAc were more prone to collapse thanin those prepared with NMP, although the membrane prepared with NMP were completely blocked at a low level of glycerol time (no glycerol post-treatment). The interaction of DE, presented in Appendix A, shows that the effect of glycerol time at a low level of the nonsolvent ratio (E) is more significant than that at a high level. As can be seen, when there is no glycerol post-treatment, the presence of isopropanol in the coagulation bath plays a decisive role by increasing the flux.

On the contrary, when there is glycerol post-treatment, the isopropanol plays a negative role and decreases the fluxes. Therefore, there should be a potential balance point between isopropanol and glycerol. However, since the glycerol post treatment could improve the flux regardless of level factor E, the positive role of isopropanol in the absence of glycerol was sacrificed in favor of glycerol post-treatment.

A regression model was applied to the base of all factors and significant interactions to find an approximation of factors for optimization. The resulting fitted model is:*Avrage Flux* = 16.7 − 0.803 *X*_A_ + 0.62 *X*_B_ + 1.76 *X*_C_ + 138 *X*_D_ + 1.98 *X*_E_ − 0.0266 *X*_A_*X*_E_ − 0.837 *X*_D_*X*_E_(1)
where, the variable *X*i represents values of A, B, C, D, and E, in sequence. The variables *X*_A_*X*_E_ and *X*_D_*X*_E_ represent the AE and DE interactions, respectively. To maximize the flux in the above equation, the variables with a positive coefficient and those with a negative coefficient should be set at a high and a low level, respectively. Therefore, the factors of B and C were set at a high level, but other factors could not be easily set due to their interactions. As mentioned above, a strong positive effect of glycerol post-treatment was identified on the flux. Hence, factor D was also set to a high level. The two-way ANOVA was applied not only to adjust the remaining two factors (i.e., A and E), but also to narrow down optimization.

The normal probability plot of the main effects on the top layer thickness is shown in Appendix A. The graph shows that the two factors of A (solvent ratio) and E (nonsolvent ratio) were significant while others were not. The plots of these two main effects (Appendix A) indicate that both have almost the same kind of effect on the top layer because of their similar slopes, and also that there is no curvature since the center points are close to lines. However, further investigations were conducted with two-way ANOVA to analyze the actual effects.

### 3.2. Two-Way ANOVA

A two-way analysis of variance (ANOVA) was performed on the two factors of A and E to find the optimum adjustment and the best combination for membrane fabrication. Each factor was set at multiple levels between high (+) and low (−), and the responses were measured at each intersection of factors. By using this method, the effect of each factor (main effects), as well as any interactions between the factors, can be estimated [24]. Table 3 is the ANOVA Table for factor A at three levels and E at five levels with the response of flux. The results were analyzed by Minitab Software to find the significance of each factor. The normality assumption was checked by constructing a normal probability plot of residuals. A normal probability plot of residuals is shown in Appendix A. Because the residuals lie approximately along a straight line, there is no problem with normality in the data. The analysis of variance for fluxes shows that the P-values for both effects were lower than 0.05, meaning both factors were significant.

Appendix A shows the individual value plot of flux versus factors A and E. In the case of pure DMAc or a mixture of NMP and DMAc, flux decreased with an increase in the concentration of isopropanol in the coagulation bath. In the case of pure NMP as the solvent, the flux reached a peak and then declined. The flux versus solvent ratio for various isopropanol concentrations is shown in Figure 1. The graph clearly shows that the maximum flux was achieved by choosing the DAMc as the solvent and pure water as a nonsolvent. However, in the case of NMP as the solvent, maximum flux was achieved with 50% isopropanol as the nonsolvent.

Figure 2 shows the SEM images of cross-sections of PES membranes prepared with DMAc, DMAc/NMP (50/50), and NMP as the solvent. As shown in Figure 2, the membrane prepared with DMAc was very porous, resulting in higher flux. This high porosity was observed everywhere in the membrane, including a dense layer in the upper part and finger-like pores in the lower part of the membrane. The solubility parameter between solvent and nonsolvent is an important parameter in membrane formation. If the difference between the solubility of solvent and nonsolvent is low, the counter diffusion of solvent in a coagulation bath and the nonsolvent in a polymer film is easier, resulting in instantaneous demixing and, usually, the formation of a porous top layer and finger-like pores in the support layer. In contrast, if the difference between the solubility of solvent and nonsolvent is high, delayed demixing occurs, leading to the formation of a dense layer [26,27]. Since the difference between solubility parameters of DMAc and nonsolvent was smaller than that between NMP and nonsolvent, the porous structure decreased with an increase in the concentration of NMP in the casting solution (as the porous structure decreased between Figure 3a,c). L. Wu et al. [19] investigated the effect of different solvents (dimethyl sulfoxide (DMSO), NMP, DMAc, and *N*,*N*-dimethylformamide (DMF)) on the performance of the membranes. They found that solubility and polarity parameters of solvent affect flux, BSA retention, and shrinkage ratio. Among the various solvents, DMAc had the highest pure water flux, and the lowest BSA retention and shrinkage ratio.

#### 3.2.1. Effect of Nonsolvent (Factor E)

Cross-sections of membrane structures formed by immersion of PES-NMP solutions in mixtures of water and isopropanol are shown in Figure 3. As can be seen, the microvoids walls become thinner, and the interconnectivity of pores increases. However, significant changes in the top layer thicknesses were not observed (see Figure 4). Figure 4 also shows that the increase of isopropanol in the coagulation bath results in a suppression of the macrovoids and the development of a thigh support layer in the lower part of the membrane, leading to a decline in flux [4,28].

The effect of the presence of isopropanol in the coagulation bath for the PES-DMAc membrane is shown in Figure 5. The length and width of macrovoids change for various nonsolvent mixtures [29]. If the isopropanol content is increased in the coagulation bath, the number of microvoids and the relative length of the microvoids decrease. By increasing the concentration of isopropanol in the coagulation bath, the top layer thickness also increases dramatically and leads to a rapid flux decline (Appendix A and Figure 4). According to S.P. Deshmukh et al. [30], by adding ethanol into the nonsolvent medium, the diffusion rate of solvent on nonsolvent during phase separation is significantly reduced. Moreover, less porosity and the shorter finger-like structure in the PVDF hollow fiber membrane were investigated by increasing ethanol in the water bath (Figure 5).

#### 3.2.2. BSA Rejection

One of the concerns for adjusting the remaining two parameters, i.e., A and E was the fouling resistance. This means that the factors were set so that the maximum flux was considered while the fouling resistance was ignored. Therefore, a BSA filtration was applied to investigate the fouling resistance of the selected membranes. Figure 6a,b shows the flux and rejection of the BSA solution (0.5 wt.%) for the PES-DMAc and the PES-NMP membranes for various nonsolvent ratios.

As expected, the maximum flux for the PES-DMAc membrane was achieved with pure water as a coagulation bath while the maximum rejection was observed in the presence of 30% isopropanol in the coagulation bath. Hence, for the preparation of the PES-DMAc membrane, factor E should be set at 30% isopropanol. Figure 6b shows that both flux and rejection were high at 70% isopropanol concentration, although the maximum pure water flux was observed at 50% isopropanol (Appendix A). Therefore, for the preparation of the PES- DMAc membrane, it is recommended to set factor E at 70% isopropanol.

In Figure 6b, both flux and rejection are high at 70% isopropanol concentration, although the maximum pure water flux is observed at 50% isopropanol (Appendix A). Therefore, for the preparation of the PES-DMAc membrane, it is recommended that factor E be set at 70% isopropanol. However, 30% of isopropanol decreased the rejection by increasing the flux, which was insignificant. Hence, the minimum percentage of isopropanol must be at least 50% to obtain the desired performance. It is highly recommended to employ 70% isopropanol to reach the water flux of 42 LMH with approximately 99% of BSA rejection.

### 3.3. Effect of Template Agents

#### BSA Rejection

One of the methods for determining the fouling status is to consider the trend of BSA rejection. Figure 7a illustrates the fouling trends of neat nanocomposite TiO_2_ membrane plus modified membranes via F127, PEG, and IM22 template agents. A significant difference is observed between the control sample and the modified membranes after 100 min, and therefore the fouling test was stopped within two hours. With regard to fouling, The F127 templating agent contributory factor caused the most resistant membrane, while IM22 and PEG were represented as the second and third antifouling properties, respectively. Figure 7b demonstrates the flux and rejection of the BSA solution (0.5 wt.%) for the nanocomposite TiO_2_ membranes with F127, PEG, and IM22 template agents. As can be seen, the maximum flux occurred for F127-modified nanocomposite membrane, and the fouling of all modified membranes was lower than that of the neat membrane. Although the pure water flux of the control membrane was higher than that of other membranes (Figure 7b), its flux recovery was lower than other composite membranes.

Water flux and flux recovery of the neat and modified membranes are shown in Appendix A. Two parameters play a role regarding the flux properties of modified membranes. First, the initial flux was declined dramatically (77%) for the sample modified via F127. However, the flux recovery of 83.33% made this sample competitive. Initial pure water flux of the specimen modified with IM22 was reasonable (3000 LMH). Subsequently, IM22 exhibited a flux recovery of 73.33%. Hence, modification using IM22 is suggested by this study.

In summary, this research performed an investigation of different parameters on membrane fabrication and various parameters on the fabrication of a nanocomposite PES/TiO_2_ membrane. Fabrication variables, including exposure time [31], immersion time [32], the solvent ratio (ratio of less volatile solvent (NMP) to more volatile solvent DMAc), glycerol time (pore filling time), and the nonsolvent ratio (isopropanol to water) were set individually. According to the analysis of the two-level factorial design, glycerol time, solvent ratio, and nonsolvent ratio were the most important factors which should be set accurately.

## 4. Conclusions

The results presented here show that the fraction factorial design along with two-way ANOVA could effectively identify influential parameters and their main effects and interactions for the preparation of PES membranes. Fabrication variables including exposure time, immersion time, the solvent ratio (ratio of less volatile solvent (NMP) to more volatile solvent DMAc), glycerol time, and the nonsolvent ratio (isopropanol to water) were set individually. According to the analysis of the two-level factorial design, glycerol time, solvent ratio, and nonsolvent ratio were the most important factors which should be set accurately. The results for the optimization of PES nanocomposite membrane fabrication are summarized as follows: an exposure time of 60 s, an immersion time above 10 h, a glycerol time of 4 h, and a nonsolvent ratio (isopropanol/water) of 30/70 for PES-DMAc membrane and 70/30 for PES-NMP membrane. In addition, IM22 caused initial pure water flux of 3000 LMH, flux recovery of 73.33%, and antifouling. Regarding these contributory factors, IM22 modification is recommended.

## Figures and Tables

**Figure 1 molecules-24-02924-f001:**
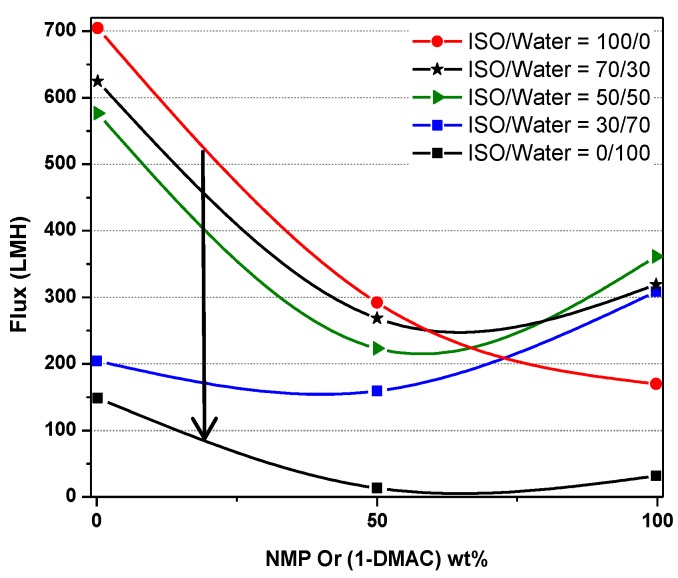
Flux versus solvent concentration for various isopropanol concentrations.

**Figure 2 molecules-24-02924-f002:**
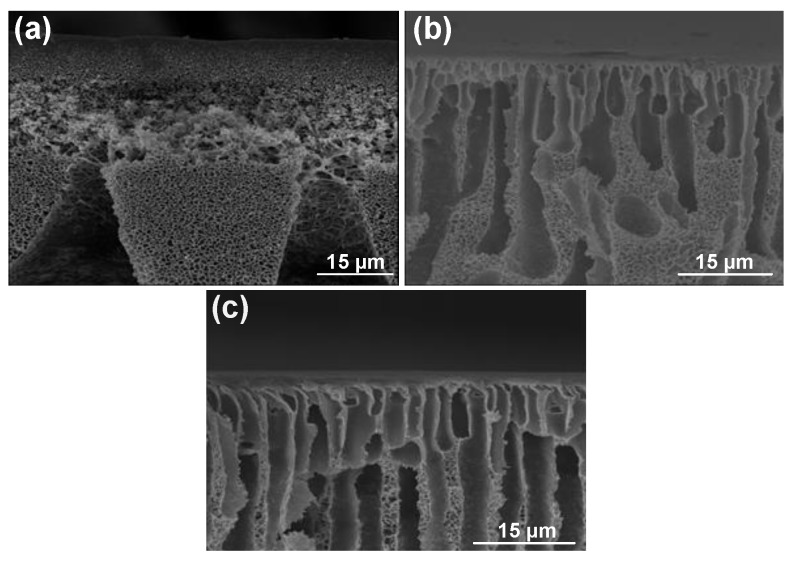
The effect of solvent on polyether sulfone (PES) membrane structure: (**a**) dimethylacetamide (DMAc), (**b**) N-methyl-2-pyrrolidone (NMP)/DMAc (50/50), and (**c**) NMP.

**Figure 3 molecules-24-02924-f003:**
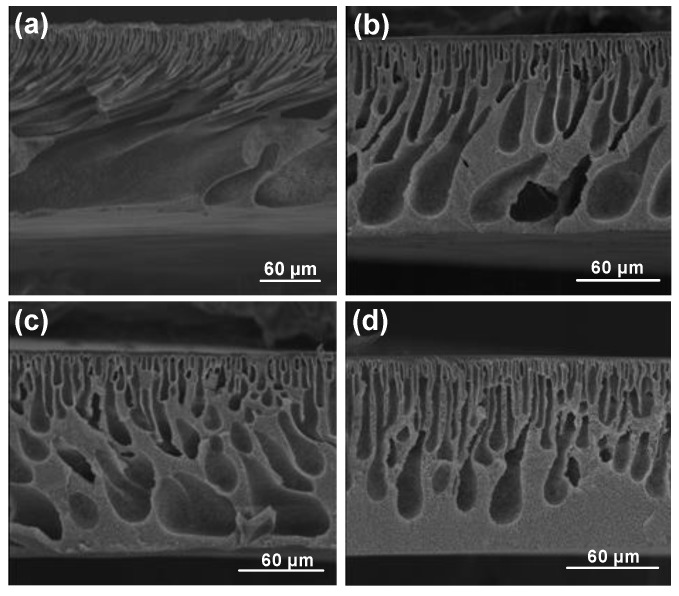
Membrane prepared with NMP as the solvent by coagulation into mixtures of water and isopropanol (concentration of isopropanol indicated): (**a**) 0% Iso, (**b**) 50% Iso, (**c**) 70% Iso, and (**d**) 100% Iso.

**Figure 4 molecules-24-02924-f004:**
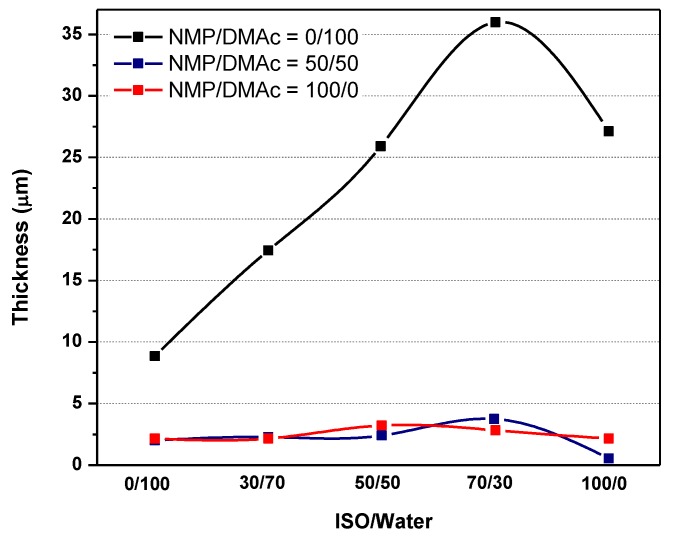
Membrane top layer thickness versus nonsolvent ratio for three different solvent ratios.

**Figure 5 molecules-24-02924-f005:**
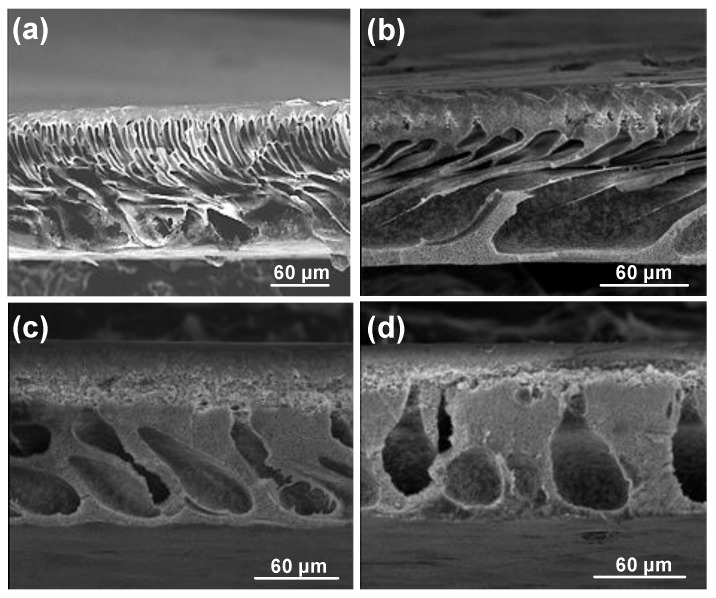
Membrane prepared with DMAc as the solvent by coagulation into mixtures of water and isopropanol (concentration of isopropanol indicated): (**a**) 0% Iso, (**b**) 50% Iso, (**c**) 70% Iso, and (**d**) 100% Iso.

**Figure 6 molecules-24-02924-f006:**
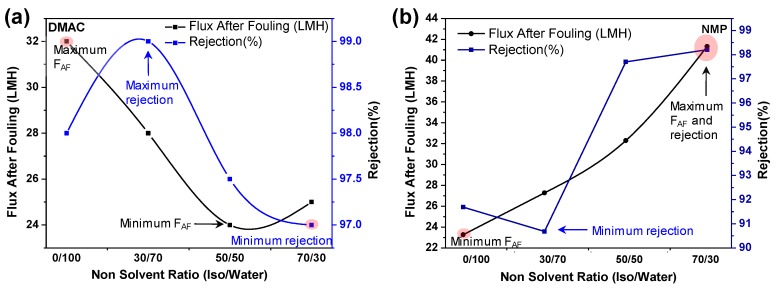
The effect of choosing various nonsolvent ratios on the flux after fouling (F_AF_) and rejection of (**a**) PES-DMAc membrane (filtration of bovine serum albumin (BSA) 0.5 wt.%) and (**b**) PES-NMP membrane (filtration of BSA 0.5 wt.%).

**Figure 7 molecules-24-02924-f007:**
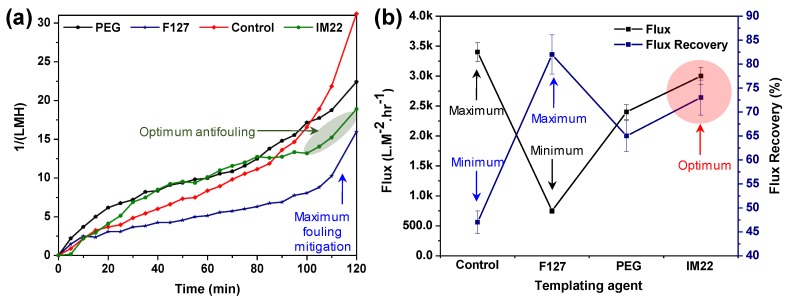
(**a**) The fouling trends of neat nanocomposite TiO_2_ membrane plus modified membranes via F127, PEG, and IM22 template agents and (**b**) water flux and flux recovery of the modified and control membranes.

**Table 1 molecules-24-02924-t001:** One-half fractional factorial design (response: flux).

Run	Treatment Combination					Level of Factors (High Level:+; Low Level: −)					Response: Flux (LMH)			
											Replicates		Average	
	Aa	Bb	Cc	Dd	Ee	Aa	Bb	Cc	Dd	Ee	i	ii	iii	
1	100	0	1	0	0	+	−	−	−	−	0	0	0	0
2	100	0	1	4	100	+	−	−	+	+	0	0	0	0
3	0	0	16	0	0	−	−	+	−	−	4	3.86	4.5	4.12
4	0	60	16	0	100	−	+	+	−	+	180	225	160	188
5	0	0	16	4	100	−	−	+	+	+	540	520	510	523
6	0	0	1	4	0	−	−	−	+	−	580	556	568	568
7	0	60	1	4	100	−	+	−	+	+	580	640	610	610
8	100	60	16	4	100	+	+	+	+	+	9	8.5	8	8.5
9	100	60	16	0	0	+	+	+	−	−	0	0	0	0
10	0	0	1	0	100	−	−	−	−	+	72	82.5	76	76.8
11	0	60	1	0	0	−	+	−	−	−	0	0	0	0
12	0	60	16	4	0	−	+	+	+	−	740	686	680	702
13	50	30	8.5	2	50	0	0	0	0	0	343	390	367	367
14	100	0	16	0	100	+	−	+	−	+	0	0	0	0
15	100	60	1	0	100	+	+	−	−	+	0	0	0	0
16	100	60	1	4	0	+	+	−	+	−	455	461	450	455
17	100	0	16	4	0	+	−	+	+	−	480	509	501	497

A: solvent ratio (weight percent ratio); B: exposure time (second); C: immersion time (hours); D: glycerol time (hour); E: nonsolvent ratio (volumetric ratio).

**Table 2 molecules-24-02924-t002:** One-half fractional factorial design (response: top layer thickness).

Run	Treatment Combination					Level of Factors (High Level: +; Low Level: −)					Average of Membrane Top Layer Thickness (µm)
	Aa	Bb	Cc	Dd	Ee	Aa	Bb	Cc	Dd	Ee	
1	100	0	1	0	0	+	−	−	−	−	2.65
2	100	0	1	4	100	+	−	−	+	+	0
3	0	0	16	0	0	−	−	+	−	−	4.3
4	0	60	16	0	100	−	+	+	−	+	2
5	0	0	16	4	100	−	−	+	+	+	1.12
6	0	0	1	4	0	−	−	−	+	−	8
7	0	60	1	4	100	−	+	−	+	+	2
8	100	60	16	4	100	+	+	+	+	+	0.75
9	100	60	16	0	0	+	+	+	−	−	1.2
10	0	0	1	0	100	−	−	−	−	+	2
11	0	60	1	0	0	−	+	−	−	−	3.2
12	0	60	16	4	0	−	+	+	+	−	9
13	50	30	8.5	2	50	0	0	0	0	0	2.5
14	100	0	16	0	100	+	−	+	−	+	1.5
15	100	60	1	0	100	+	+	−	−	+	0.18
16	100	60	1	4	0	+	+	−	+	−	1.8
17	100	0	16	4	0	+	−	+	+	−	2.15

A: solvent ratio (weight percent ratio); B: exposure time (second); C: immersion time (hours); D: glycerol time (hour); E: nonsolvent ratio (volumetric ratio).

**Table 3 molecules-24-02924-t003:** Two-way analysis of variance (ANOVA): Flux versus solvent ratio (A) and the nonsolvent ratio (E).

NMP/DMAc (A)	ISO/Water (E)				
	0/100	30/70	50/50	70/30	100/0
0/100	702	622	582	204	146.5
50/50	290	265	224	158	8
100/0	268	319	360	310	27

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
