# Peer review of "ANOVA Design for the Optimization of TiO_2_ Coating on Polyether Sulfone Membranes"

_molecules, 2019, doi:10.3390/molecules24162924_

Round 1

Reviewer 1 Report

 ANOVA design for optimization of poly(ether sulfone) membrane fabrication

1-   The supplementary was not submitted!! Without it, the review is not possible.

2-   The relation of this paper with this journal should be clarified. This manuscript is more appropriate for membranes journals.

3-   The novelty is not clear. some researches published before for optimization of PES and other nanocomposite membranes. 

4-   The abstract should be revised. More details of the results should be added.

5-   The introduction should be revised. More related research should be added and connect to the aim of this research.

6-   The general concepts should be deleted.

7-   Where is FigA.1 to FigA.9? The supplementary was not submitted?

8-   Table 3 should be addressed in the manuscript text.

9-   The SEM images have low quality. They should be changed with better resolution.

10-The trend of results presentation is confusing. It should be revised.

Author Response

Responses to Reviewer’s Comments

Journal: Molecules

Manuscript ID: molecules-550638

Title: " ANOVA design for optimization of TiO2 coated poly(ether sulfone) membrane "

Author(s): Yasin Orooji*; Ehsan Ghasali; Nahid Emami; Fatemeh Nourisfa; Amir Razmjou

We highly appreciate the reviewer for the detailed and instructive suggestions. Thanks for your time and efforts. Suggestions made by the reviewers are really helpful to improve the quality of our present work. We have done our best to comply with them in this thoroughly revised manuscript. Followings are our responses to the reviewers’ comments.

Reviewer #1:

Special thanks to Reviewer #1 for his/her valuable comments.

Comment 1

The supplementary was not submitted!! Without it, the review is not possible.

Response to comment 1

Thanks for the comment; the appendix was added.

Comment 2

The relation of this paper with this journal should be clarified. This manuscript is more appropriate for membranes journals.

Response to comment 2

We agree that this paper can also be submitted to membranes; however, both journals have some overlaps, which made the option for us to select Molecules instead of Membranes. In this Special Issue, researchers are invited to submit their original research articles on synthesis of different materials engineered for environmental applications. Scientists are encouraged to publish their experimental and theoretical results in as much detail as possible in Molecules. Molecular assembly effect of different organic molecules on the morphology of TiO2 nanoparticle clusters are formed on the membrane surface. Interaction between templating agents and TiO2 crystals at a molecular level dictates how TiO2 crystals nucleate and grow.

Comment 3

The novelty is not clear. some researches published before for optimization of PES and other nanocomposite membranes.

Response to comment 3

The authors greatly appreciate this valuable comment.

The novelty of this work lies in materials design parameters that have been adjusted by ANOVA analysis using different templating agents. In addition, all the previous optimization research works were focused on the optimization of mixed matrix membranes that the nanoparticles were used as the fillers. However, in this work, nanoparticles co-operated on the surface of the membranes. The incorporation of nanoparticles into polymeric membranes falls into two categories: blending (melt, solvent, sol-gel mixing and in-situ grafting) and coating (direct deposition and low-temperature hydrothermal process). In the former technique, the nanoparticles are dispersed into the flesh of the membranes. Achieving a homogenous uniform composite wherein the nanoparticles are uniformly dispersed is the major challenge in the blending approach. In the latter approach, the nanoparticles are deposited and coated on the top surface of the membranes via solution adsorption or vacuum filtration or spin-coating or dip-coating. The most important issue is achieving a uniform coating with high stability and durability of the coated particles.

The following sentences were added to the abstract:

The optimized parameters obtained in this study are the exposure time: 60 seconds; immersion time: above 10 hr; glycerol time: 4 hr; and the non-solvent volumetric ratio (isopropanol/water): 30/70 for PES- DMAc membranes and 70/30 PES-NMP membranes.

The following paragraph was added to the introduction:

“In this study, coating layers which provide optimum preparation conditions for phase separation process has been studied. This study is based on the performance of PES membrane in water treatment by investigating different parameters in the membrane fabrication and various parameters in the fabrication of nanocomposite PES/TiO2 membrane. The results presented here show that the fraction factorial design along with the two-way ANOVA could effectively be useful to identify important parameters and their main effects as well as interactions for the preparation of PES membranes. ”

The following paragraph was added to the discussion:

“Fabrication variables, including exposure time, immersion time, the solvent ratio (ratio of less volatile solvent (NMP) to more volatile solvent DMAc), glycerol time (pore filling time), and the non-solvent ratio (isopropanol to water) were set separately. According to the analysis of two-level factorial designs, glycerol time, solvent ratio, and non-solvent ratio were the most important factors, which should be set accurately.

Moreover, Figs. 8  9 were similarly corrected to response to the raised point as shown in the following:

Comment 4

The abstract should be revised. More details of the results should be added.

Response to comment 4

As suggested, some sentences were corrected in the abstract as  follows:

“The optimized parameters obtained in this study are the exposure time: 60 seconds; immersion time: above 10 hr; glycerol time: 4 hr; and the non-solvent volumetric ratio (isopropanol/water): 30/70 for PES-DMAc membranes and 70/30 PES-NMP membranes. Comparison of contributory factors of different templating agents.”

Comment 5

The introduction should be revised. More related research should be added and connect to the aim of this research.

Response to comment 5

The introduction has been revised.

Comment 6

The general concepts should be deleted.

Response to comment 6

We highly appreciate the reviewer’s positive comments and instructive suggestions and did our best to omit the general concepts and present them in the appendix.

Comment 7

Where is FigA.1 to FigA.9? The supplementary was not submitted?

Response to comment 7

Thanks for the comment; the appendix was added.

Comment 8

Table 3 should be addressed in the manuscript text.

Response to comment 8

We highly appreciate the reviewer for raising the point. Thanks for the comments, the table was addressed in the context accordingly.

“Table 3 is the ANOVA Table for factor A in 3 levels and E in 5 levels with the response of flux.”

Comment 9

The SEM images have low quality. They should be changed with better resolution.

Response to comment 9

Thanks for the comments, the images were adjusted via Adobe Photoshop accordingly to address the comments. Moreover, the photos are uploaded as  Visio Files for further corrections (the original images with the highest quality obtained from the SEM are provided in the Visio Files).

Comment 10

The trend of results presentation is confusing. It should be revised.

Response to comment 10

The authors greatly appreciate this thoughtful comment. To address the reviewer's comment, the illustrations were improved as mentioned in response to comments 3 and 9;  the related context and captions were fixed accordingly.

Reviewer 2 Report

Reviewer’s comments:

Orooji et al., fabricated poly (ether sulfone) (PES) membranes and optimize its antifouling and the mechanically stable surface by Anova. This is very well written paper. But still some necessary improvement is needed. The observed quarries are:

1.      Title can be changed. Please check it.

2.      Abstract should be quantitative and F127, PEG, and IM22 17 template agents should be avoided. Please check it.

3.      The digital image of membrane and Template agents should be inserted. A scheme for preparation of membrane will be helpful for reader. Just try to insert it.

4.      The information regarding used TiO2 is less. Please improve it in introduction. Cite this paper. Applied Catalysis A: General 425, 110-116

5.      Line 259……“The figure also shows that the increase of isopropanol  in the coagulation bath results in a suppression of the macro-voids and development of thigh support  layer in the lower part of the membranes, which leads to a decline in flux”………needs proper evidence. Better check some paper.

6.      Conclusion part should be more informative. Just check it.

7.      Line…….304 …..3.3.1………“BSA rejection”………This paragraph should be confirmed by some papers. Example….”Although the pure water flux of the control 315 membrane was higher than that of other membranes (see Figure 9), its flux recovery is lower than 316 other composite membranes”……..needs some references.

Author Response

Responses to Reviewer’s Comments

Journal: Molecules

Manuscript ID: molecules-550638

Title: " ANOVA design for optimization of TiO2 coated poly(ether sulfone) membrane "

Author(s): Yasin Orooji*; Ehsan Ghasali; Nahid Emami; Fatemeh Nourisfa; Amir Razmjou

We highly appreciate the reviewer for the detailed and instructive suggestions. Thanks for your time and efforts. Suggestions made by the reviewers are really helpful to improve the quality of our present work. We have done our best to comply with them in this thoroughly revised manuscript. Followings are our responses to the reviewers’ comments.

Reviewer #2:

Special thanks to Reviewer #1 for his/her valuable comments.

Orooji et al., fabricated poly (ether sulfone) (PES) membranes, optimized its antifouling and the mechanically stable surface by ANOVA. This is a very well-written paper. But still, some improvements are needed. The observed quarries are:

Comment 1

Title can be changed. Please check it.

Response to comment 1

Thank you for your suggestion. The title was changed to “ANOVA design for optimization of TiO2 coated poly(ether sulfone) membrane”.

Comment 2

Abstract should be quantitative and F127, PEG, and IM22 17 template agents should be avoided. Please check it.

Response to comment 2

As suggested, some sentences were corrected in the abstract as follows:

The optimized parameters obtained in this study are the exposure time: 60 seconds; immersion time: above 10 hr; glycerol time: 4 hr; and the non-solvent volumetric ratio (isopropanol/water): 30/70 for PES-DMAc membranes and 70/30 PES-NMP membranes. Comparison of contributory factors of different templating agents with the control nanocomposite membrane revealed that F127 triblock copolymer resulted in an excellent antifouling membrane with a higher bovine serum albumin rejection and flux recovery of 83.33%.

Comment 3

The digital image of membrane and Template agents should be inserted. A scheme for preparation of membrane will be helpful for reader. Just try to insert it.

Response to comment 3

The authors greatly appreciate this valuable comment. The following figure was added to appendix and addressed in the context:

Comment 4

The information regarding used TiO2 is less. Please improve it in introduction. Cite this paper. Applied Catalysis A: General 425, 110-116

Response to comment 4

The proper paper suggested was added to the introduction .

“Although it has been reported that TiO2 nanoparticles concentration within PES and PVDF matrices using phase inversion approach is about 2 wt %, there is still a discrepancy in the literature when it comes to TiO2 coating on PES membranes.15, 16

15.       M. Yadav, D. K. Mishra and J.-S. Hwang, Applied Catalysis A: General, 2012, 425-426, 110-116.

16.       S. Meng, J. Mansouri, Y. Ye and V. Chen, J Membrane Sci, 2014, 450, 48-59.

Comment 5

Line 259……“The figure also shows that the increase of isopropanol in the coagulation bath results in a suppression of the macro-voids and development of tight support layer in the lower part of the membranes, which leads to a decline in flux”………needs proper evidence. Better check some paper.

Response to comment 5

The figure also shows that an increase in isopropanol in the coagulation bath results in the suppression of the macro-voids and development of thigh support layer in the lower part of the membranes,  leading to a decline in flux 10

10. A. C. S. Razmjou Chaharmahali and F. o. E. U. Engineering, Journal, 2012.

Comment 6

Conclusion part should be more informative. Just check it.

Response to comment 6

Yes, we agree with this comment.

The results could be summarized as follows:  the exposure time of 60 s, immersion time of above 10 hrs, glycerol time of 4 hrs. and the nonsolvent ratio (isopropanol/water) of 30/70 for PES-DMAC membranes and 70/30 PES-NMP membranes for the optimization of PES nanocomposite membranes fabrication. IM22 caused initial pure water flux of 3000 LMH, flux recovery of 73.33% and antifouling. Regarding these contributory factors, IM22 modification is recommended.

We also added the following paragraph before conclusion:

Fabrication variables, including exposure time, immersion time, the solvent ratio (ratio of less volatile solvent (NMP) to more volatile solvent DMAc), glycerol time (pore filling time), and the non-solvent ratio (isopropanol to water) were set separately. According to the analysis of two-level factorial designs, glycerol time, solvent ratio, and non-solvent ratio were the most important factors, which should be set accurately.

Comment 7

Line…….304 …..3.3.1………“BSA rejection”………This paragraph should be confirmed by some papers. Example….”Although the pure water flux of the control 315 membrane was higher than that of other membranes (see Figure 9), its flux recovery is lower than 316 other composite membranes”……..needs some references

Response to comment 7

Thanks for the thoughtful comments; all of these comparisons were done based on the results of this manuscript. Hence, Figs. 8 and 9 were merged and some points were highlighted to increase the clarity of context, as presented below:

Round 2

Reviewer 1 Report

The authors have responded to my comments. 

Reviewer 2 Report

Author is well addressed my queries. So I recommended it for final publication.